# Grazing Sheep Behaviour Recognition Based on Improved YOLOV5

**DOI:** 10.3390/s23104752

**Published:** 2023-05-15

**Authors:** Tianci Hu, Ruirui Yan, Chengxiang Jiang, Nividita Varun Chand, Tao Bai, Leifeng Guo, Jingwei Qi

**Affiliations:** 1College of Computer and Information Engineering, Xinjiang Agricultural University, Urumqi 830052, China; 15036893998@163.com (T.H.);; 2Agricultural Information Institute of Chinese Academy of Agricultural Sciences, Beijing 100081, China; 3Institute of Agricultural Resources and Regional Planning, Chinese Academy of Agricultural Sciences, Beijing 100081, China; 4College of Agriculture, Fisheries and Forestry, Fiji National University, Suva P.O. Box 7222, Fiji; 5Xinjiang Agricultural Information Technology Research Centre, Urumqi 830052, China; 6Ministry of Education Engineering Research Centre for Intelligent Agriculture, Urumqi 830052, China; 7College of Animal Sciences, Inner Mongolia Agricultural University, Hohhot 010018, China

**Keywords:** improved YOLOV5, pasture, grazing sheep, behaviour recognition

## Abstract

Fundamental sheep behaviours, for instance, walking, standing, and lying, can be closely associated with their physiological health. However, monitoring sheep in grazing land is complex as limited range, varied weather, and diverse outdoor lighting conditions, with the need to accurately recognise sheep behaviour in free range situations, are critical problems that must be addressed. This study proposes an enhanced sheep behaviour recognition algorithm based on the You Only Look Once Version 5 (YOLOV5) model. The algorithm investigates the effect of different shooting methodologies on sheep behaviour recognition and the model’s generalisation ability under different environmental conditions and, at the same time, provides an overview of the design for the real-time recognition system. The initial stage of the research involves the construction of sheep behaviour datasets using two shooting methods. Subsequently, the YOLOV5 model was executed, resulting in better performance on the corresponding datasets, with an average accuracy of over 90% for the three classifications. Next, cross-validation was employed to verify the model’s generalisation ability, and the results indicated the handheld camera-trained model had better generalisation ability. Furthermore, the enhanced YOLOV5 model with the addition of an attention mechanism module before feature extraction results displayed a mAP_@0.5_ of 91.8% which represented an increase of 1.7%. Lastly, a cloud-based structure was proposed with the Real-Time Messaging Protocol (RTMP) to push the video stream for real-time behaviour recognition to apply the model in a practical situation. Conclusively, this study proposes an improved YOLOV5 algorithm for sheep behaviour recognition in pasture scenarios. The model can effectively detect sheep’s daily behaviour for precision livestock management, promoting modern husbandry development.

## 1. Introduction

Recently, animal welfare has been discussed alongside intensive farming [1,2] for livestock species such as cattle, sheep, and horses, linking natural pasture benefits to animal health. Nevertheless, complications occur in raising livestock [3], especially in constructing modern natural pastures and sustaining grassland resources which consequently influence animal welfare. Likewise, it is vital to design supporting modern monitoring equipment. Since animal behaviour is a good indicator of animal health, monitoring animal behaviour makes a good foundation for the timely detection of abnormalities. Despite that, manual observation makes it impossible to meet the schedule in pastures where animals are spatially active while being inefficient and costly when numerous animals are observed [4]. Hence, the scenario constitutes the essentiality of using certain monitoring equipment to observe the daily animal behaviour in specific situations (e.g., watering holes, etc.). In other words, introducing automatic identification for analysing animal behaviour would benefit animal well-being in open pastures while improving the efficiency of farm monitoring [5].

Presently, two predominant approaches in animal monitoring are: contact sensor devices and video surveillance-based camera devices. Sensor devices typically consist of: collars, cage covers, ear tags, and pedometers. These devices serve functions as positioning, temperature monitoring, acceleration, and sound sensors. The video surveillance devices are primarily camera-based computer vision methods. Here, the video data are combined with deep learning models for behavioural recognition, abnormal state recognition, and other observations of the animals’ daily behaviour. As a non-contact and cost-effective technology, computer vision is a prominent trend in animal behaviour recognition [1,6,7,8,9,10,11]. Computer vision is consistently evident in research on smart farming in animal husbandry for pig, cattle, and sheep species, detecting behaviours that have led to crucial indicators for solving real-life production problems.

Researchers such as Nasirahmadi et al. have used captured images for processing to study the lying patterns of pigs under commercial farm conditions [12,13]. Meanwhile, Chen et al. [14] conceptualised a ResNet50 convolutional neural network and long short-term memory (LSTM)-based method for pig drinking and water dispenser play behaviour recognition, with classification accuracies of 0.87 and 0.93 in the body and head regions of interest, respectively. Jiang et al. [15] proposed an algorithm based on YOLOV3 for critical part detection in dairy cows in complex scenarios, and the algorithm detected each part with an average accuracy of 0.93. Similarly, Cheng et al. [16] proposed a deep YOLOV5 network-based behaviour recognition model for sheep in housed scenes, showing that the algorithm can be used in structured settings with a deep learning model in a structured scenario. This exhibits that it is unnecessary to use a large amount of training data when the training data and the data generated in a real application have the same features. At the core of computer vision are the deep learning algorithms used for detection, the most important of which are convolutional neural networks (CNNs), capable of automatically learning invariant features in a task in a layered manner [17]. Faster region-convolutional neural networks (R-CNNs) [18] and ZFnet [19] were used to identify the individual feeding behaviour of pigs [20] by detecting the letters marked on the head. In surveillance videos, Faster R-CNN was used to detect dairy goats [21]. 

Moreover, Wang et al. modified YOLOV3 based on the filter layer, which was employed to detect key parts of cows in natural scenes [14,22]. Wen et al. [23] applied a convolutional neural network (CNN) for a cow detection and counting system, obtaining an accuracy of 0.957. A YOLOV5-ASFF target detection model to determine bovine body parts, namely, individual, head, and leg, is proposed by Qiao et al. [24] in complex scenes. The target detection model is based on convolutional neural networks for feature extraction and is a contemporary computer vision method applied in all fields of precision agriculture. For instance, Zhao et al. [25] utilised an improved YOLOV5-based method to accurately detect wheat spikes in unmanned aerial vehicle (UAV) images to address false detection and missed detection of spikes caused by occlusion conditions. Wang et al. [26] built an image acquisition system based on a fruit pose adjustment device and used the deep learning-based YOLOV5 algorithm for the real-time recognition of apple stems/calyxes.

A Faster R-CNN model based on the Soft-NMS algorithm was used for real-time detection and localisation of sheep under complex feeding conditions [27]. The current literature on sheep behaviour particularly refers to studies with intensive housing conditions, which make observation and data collection more accessible and convenient. Yet, fewer studies are related to computer vision applications under natural grazing conditions. The reasons could be that the large size of the pasture is not conducive to the application, and the density of sheep in free grazing conditions is too high, consequently placing greater demand on data acquisition and the robustness of subsequent models. To counter the challenge of monitoring sheep behaviour in pastures, an improved YOLOV5 recognition algorithm is proposed in this study to detect sheep’s daily behaviour (standing, feeding, and lying) in grazing pastures. At the same time, datasets were constructed for different shooting methods to verify the model’s generalisation capability and provide a basis for subsequent applications. This research has collected relevant data in conjunction with actual pasture scenarios, focusing on the behaviour of grassland-grazing sheep to design experimental studies.

## 2. Materials and Methods

### 2.1. Experiments and Data Acquisition

The site of the experiment is located in a grassland field of a small town of Xeltala, Hulunbeier, Inner Mongolia Autonomous Region of China, longitude: 120.00435, latitude: 49.34809. The field had a length of 40 m and a width of 10 m and was fenced into three small 40 m × 3.3 m areas with natural grass and watering basins, freely available for sheep feeding and drinking.

In the experiment, two movable fences were used to adjust area size, predominantly to facilitate comprehensive observation of the sheep’s behaviour during grazing using the camera, making it easier to change the experimental site after the grass vegetation was foraged. Two filming devices were selected for the experiment: a camera that was fixed to the fence (HIKVISION Fluorite Cloud H8 model) and a handheld recording device (Snapdragon S3 sports camera). The cameras were placed on both sides of the fence (Figure 1a) to capture images of the sheep’s behaviour during free grazing. The camera positions and angles were manually adjusted on both sides of the fence to observe sheep behaviour completely. 

The handheld device (Figure 1b) was operated manually with a tripod, following the sheep’s daily behaviour through image recording. The two devices had different recording specifications; the camera device recorded in MP4 format at 1920 × 1080 and 30 frames per second, and the handheld device recorded in MP4 format with a pixel size of 1920 × 1440 at 30 frames per second. The experiment was conducted from 22 August 2022 to 19 September 2022, with a total of 22 sheep observed, with two sheep being observed in the experimental area each day in rotation to ensure that each sheep was included in the image data captured by both devices. All 22 sheep in our study were from the Hulunbeier breed, comprising 13 ewes and 9 rams, aged between one and two years. The selection criteria specifically required the exclusion of ewes with lambs or in labour to ensure that the research had only comprised healthy animals. More than 500 h of valid video for the experimental data were recorded. The behaviours observed during the experiment were feeding, standing, lying, walking, and running.

### 2.2. Dataset Construction

This study focuses on constructing a sheep behaviour dataset using the pictures of 22 sheep of different perspectives and states throughout the experiment. The images in the dataset were manually selected, considering the different lighting conditions throughout the day. The photos were selected based on the lighting conditions, and some nighttime observation photos were taken by the fixed camera video. The complete image dataset encompasses sheep feeding scenes in different weather conditions in the grassland, producing a complex data scene.

The pictures of sheep behaviour were taken both via video and handheld camera. A total of 1656 pictures were captured, of which 703 pictures were taken by fixed cameras and 953 pictures were captured by handheld devices. The sheep in the dataset had three behaviours: feeding, standing, and lying (Table 1, Figure 2). We then labelled the images to produce a label file that conformed to the YOLO training format. The images and tag files were divided into a dataset of 1490 images (1101 standing tags, 1952 feeding tags, and 600 lying tags) and a validation set of 166 images (105 standing tags, 210 feeding tags, and 67 lying tags). After this, we separated the data between the stationary camera and the handheld device for comparison training to test the model’s ability on different acquisition methods and scenes and the influence of the acquisition methods and scenes on the model.

#### 2.2.1. Scenario Dataset Construction

To verify the effect of the scene on behavioural recognition, separate datasets were constructed for the scenes presented above, with photos taken by both the fixed camera and handheld camera (Table 2). The angle of view of the sheep captured by the two camera types differed, with the fixed camera capturing a top view at about 30° and the handheld camera capturing a flat view at about 180° (Figure 3).

#### 2.2.2. The Difference between Scenarios

Two primary shooting scenarios were included to construct the scenario dataset: fixed and handheld cameras. While both methods produced different shooting angles, the primary difference was the impact of natural lighting on the resulting images. For the fixed camera, we conducted 24 h of shooting throughout the day, capturing the varying effects of sunlight on the camera images (Figure 4). The scenarios were complex and diverse, resulting in significant differences between the images. In contrast, we conducted manual shooting at a closer distance with a higher proportion of sheep in the frame for the handheld camera. The impact of lighting on the images was less pronounced, resulting in fewer differences between pictures. Overall, these different shooting methods enabled us to capture various scenarios and lighting conditions, providing a comprehensive dataset for our research (Table 3).

### 2.3. Data Enhancement

To make the model training converge faster to achieve better results and obtain a robust model, we used common data augmentation, where we changed the data systematically or randomly using code. For images, common data enhancements including flipping, adjusting colours, and adding random noise were applied (Figure 5). We also used the Mosaic data enhancement method proposed in the YOLOV4 [28] paper. The basic idea of Mosaic data enhancement is to stitch multiple images together to produce a new image. These images can be from different datasets or the same dataset. The stitched images can be used to train a grazing sheep behaviour recognition model to improve the model’s generalisation ability.

### 2.4. Model Improvement Implementation

#### 2.4.1. YOLO Series Models

The You Only Look Once (YOLO) [29] family is a fast target detection model for object recognition in images or videos, which differs from traditional classification and localisation models (e.g., the R-CNN family) by predicting multiple targets in a single step. The YOLO family of models includes YOLOV1, YOLOV2, YOLOV3, etc., with the latest version currently being the YOLOV8 model. The series of models consists of three main parts: a backbone network for extracting image features, a feature fusion layer for combining different scale features for prediction, and a YOLO detection head to give multiple predictions. The primary choice in this article is the YOLOV5s model (network structure as shown in Figure 6) for improving daily behaviour recognition during sheep grazing. Three main modules were chosen to strengthen the network. Firstly, the convolutional block attention module (CBAM) [30] was included to adjust the relationship between channel and space to enhance the performance of convolutional feature extraction. Then, the idea of the bidirectional feature pyramid network (BiFPN) [31] was introduced in the feature fusion stage to configure weights on features of different scales for fusion. Lastly, the residual variant selective kernel networks-skip connect (SKNet-sc) was added. The SKNet-sc introduces the attention mechanism to the fused feature map to enhance the feature role and improve the accuracy for subsequent detection.

#### 2.4.2. CBAMs

Factoring in many lighting conditions and diverse recognition scenes to improve extract image features, we introduced the CBAM. The CBAM can enhance the ability of convolutional neural networks (CNNs) to extract features by introducing channel and spatial attention mechanisms to improve network performance.

Sanghyun et al. [30] proposed a CBAM (Figure 7) that combines feature channel information and feature spatial information, which can be divided into two submodules: the channel attention module (CAM) and the spatial attention module (SAM). The channel attention module is mainly used to adaptively adjust the importance of each channel to improve the representation of features. The spatial attention module captures the spatial relationships in the feature map. By combining the CAM and SAM, the CBAM can adaptively adjust the channels and spatial relationships to improve the performance of the CNN.

#### 2.4.3. BiFPN Network Architecture

BiFPN is a shorthand for bidirectional feature pyramid network, which is a neural network architecture commonly utilised in computer vision tasks, especially in object detection and segmentation. This architecture is an extension of the feature pyramid network (FPN) and aims to address the challenge of detecting objects at varying scales in an image. The BiFPN is made up of multiple layers that refine and aggregate features from different levels of the feature pyramid. The significant innovation of the BiFPN lies in its use of both bottom-up and top-down pathways to transmit information across layers, allowing for bidirectional flow of information. This approach captures more detailed features at different scales, resulting in superior object detection and segmentation performance. We have introduced this structure into YOLOV5; the specific network fusion structure is shown in Figure 8a. The main idea was to set different weights for different features, then add them together, and finally perform activation and convolution to obtain the fused features, as shown in Figure 8b.

#### 2.4.4. SKNet-sc Module

SKNet refers to selective kernel network, a neural network architecture developed to enhance the performance of convolutional neural networks (CNNs) in image recognition tasks. SKNet focuses on learning the importance of different convolutional kernel sizes at each network layer and selectively combining them to capture both local and global features in the input image. SKNet-sc is based on SKNet [32] with the addition of a residual connection. SKNet employs dynamically selected convolutions to improve accuracy through three main operations: split, fuse, and select. Following these operations, a skip connection is established between the output and input to integrate the fused and filtered features with the original ones. The features are then scaled up to provide a foundation for subsequent detection using the detection head. The network structure is shown in Figure 9.

#### 2.4.5. Improved YOLOV5 Model

In this study, three modules were chosen to improve YOLOV5. Before feature extraction, a CBAM was added to increase the channel and spatial attention to improve convolution performance. A subsequent convolution was performed to extract features using a BiFPN structure for feature fusion after feature extraction. The combining of features and weights of different sizes and scales to fuse features makes full use of the SKNet-sc structure for each of the three scales before detection. An attention mechanism was added to the fused features so that the feature map amplified more critical features and improved accuracy. The improved network model is shown in Figure 10.

### 2.5. Model Evaluation

This study used model evaluation metrics commonly utilised in target detection, including Precision, Recall, F1-score, average precision (AP), mean average precision (mAP), frames per second (FPS), and model size to evaluate our experimental model. Precision indicates the proportion of results predicted to be true for a behaviour of sheep; Recall indicates the proportion of all data predicted to be true for a behaviour. The F1-score is used to assess the relationship between Precision and Recall evaluations. AP is the average accuracy for each classification of sheep’s behaviour. The mAP is an average of the average accuracy of the three behavioural classifications, resulting in the average accuracy of the model as a whole, which is the main evaluation indicator of the model. At the same time, the calculation of mAP adopts the calculation method of multiple IOU thresholds. The mAP_@0.5_ is the average accuracy of the three behaviours when the IOU threshold is 0.5. The mAP_@0.75_ is the average accuracy of the three behaviours when the IOU threshold is 0.75. The mAP_@0.5:0.95_ represents the average mAP of different IOU thresholds (from 0.5 to 0.95, step size 0.05). FPS is used to evaluate the speed of object detection, which refers to the number of images that can be processed per second.

For the above model evaluation metrics, the specific formulas are as follows: true positive (TP) is the number of positive samples detected correctly. False positive (FP) is the number of negative samples detected as positive. False negative (FN) is the number of positive samples detected as negative.
$$Precision=\frac{TP}{\left(TP+TN\right)}$$
$$Recall=\frac{TP}{\left(TP+FN\right)}$$
$$F1=\frac{2\times \mathrm{Precision}\times \mathrm{Recall}}{\left(\mathrm{Precision}+\mathrm{Recall}\right)}$$
$$AP=\int_{0}^{1}P(r)dr$$
$$mAP=\frac{\sum_{i}^{K}AP_{i}}{K}$$

## 3. Results

### 3.1. Recognition Results for Different Scenes

The scenes captured by the fixed camera and the scenes captured by the handheld camera were separately processed in the YOLOV5 model for training and comparison. Different shooting methods and scenes were used to compare different effects of the recognition model on the same detection target. The model trained by the dataset of the same shooting method positively affected its own verification set, which could reach more than 90% mAP_@0.5_. The training results are shown in Table 4.

While the model performs relatively well for each scenario individually, its accuracy in recognising standing behaviour is notably lower than for the other two behaviours. This could be attributed to the fact that standing behaviour closely resembles feeding behaviour and can be easily mistaken for it. Additionally, the dataset for standing behaviour is smaller than that for feeding behaviour, which further contributes to the model’s lower accuracy in identifying standing behaviour. To verify the generalisation ability of the models in the respective scenes, cross-validation was utilised. We cross-validated the validation sets of each of the two models (using the model for the fixed camera to validate the validation set for the handheld camera and using the model for the handheld camera to validate the validation set for the fixed camera scenes). The validation results are presented in Table 5.

Through cross-validation, it can be seen that the mAP_@0.5_ of the two models decreased in contrast with the previous comparison. The generalisation ability of the fixed camera model is not as good as that of the handheld camera model, but the mAP_@0.5_ of the two models is still above 70%, indicating that they still possess a certain level of generalisation ability. To determine the effect of the model, the entire dataset was put into the model for training, and the specific results obtained are shown in Table 6.

In summary, the results indicate that the YOLOV5 model performs equally well on each dataset, but cross-validation on different datasets shows that the model generalisation ability is different. Hence, for real-world applications, data from more scenarios need to be collected to improve the robustness of the model.

### 3.2. Improved Results for YOLOV5

Before improving the model, we trained three versions of YOLOV specifically, YOLOV5, YOLOV6, and YOLOV7 on our dataset. During training, we kept the network structure of the three models constant and ensured consistent training parameters. The final training results are shown in Table 7. The results indicate that YOLOV6 and YOLOV7 did not perform well on this dataset, despite their unique network structure improvements. Thus, we ultimately chose YOLOV5 for further improvement.

To improve the recognition accuracy of the model, we refined the YOLOV5 model. We employed the CBAM before feature extraction, used BiFPN in feature fusion, and used SKNet-res to add a feature attention mechanism after feature fusion. We performed ablation experiments to explore the influence of each module on the final experimental results. Table 8 reveals the effect of each module on the experimental results in the ablation experiment. After adding the CBAM, mAP_@0.5_ increased by 0.9%, mAP_@0.75_ increased by 1.1%, mAP_@0.5:0.95_ increased by 0.4%, and the model size did not change. After replacement with the BiFPN module, mAP_@0.5_ increased by 1.1%, mAP_@0.75_ increased by 2.2%, mAP_@0.5:0.95_ increased by 0.5%, and the model’s size increased by 0.3 MB. After adding SKNet-res, mAP_@0.5_ increased by 0.4%, mAP_@0.75_ and mAP_@0.5:0.95_ decreased to varying degrees, and the model’s size increased by 110.9MB. Notably, SKNet-res has more parameters, and the effect of training after adding it is relatively poor. After that, the three modules were arranged and added to the network. Compared with the mAP_@0.5_ indicators in YOLOV5s, the indicators are improved, but the mAP_@0.75_ and mAP_@0.5:0.95_ do not show a stable improvement. The three modules were then added to the network at the same time. For all three indicators, mAP_@0.5_ increased by 1.7%, mAP_@0.75_ increased by 2.6%, and mAP_@0.5:0.95_ increased by 0.3%. Due to the SKNet-res module’s addition, the model’s size increased by 111.2 MB. In addition to evaluating the performance of the models, we analysed their inference speed using a Tesla P100 (16G) graphics card and measured the frames per second (FPS) for each model. Our results indicate that the inference speed decreased as additional modules were added, with the SKNet-sc module having the greatest impact on the performance of the models. This information is crucial for optimising the performance of the models and ensuring that they can operate efficiently in real-world scenarios.

Table 8 shows that adding a single module and combining the three modules to train the model improved the mAP_@0.5._ However, different performances are seen for mAP_@0.75_ and mAP_@0.5:0.95_. Although the three modules were added to the network structure, the model’s size increased exponentially, mainly due to the large number of calculations of SKNet-res. Table 8 illustrates that removing the SKNet module maintains the performance, as the model size is relatively small and the inference speed remains normal. From the practical engineering point of view, the final model selection includes the CBAM and BiFPN module model.

### 3.3. Model Applications

This study accomplishes the recognition of some individual sheep behavioral elements in a free ranging scenario, showing the robustness and applicability of the method. The improved YOLOV5 model is small, can be embedded in a camera after pruning for a more flexible application, and can be configured to a server for online real-time detection, facilitating the monitoring of pasture-related information. After constructing the model and studying the real-time detection system, we propose an application scenario for the model that we have tested but not yet applied in the above-mentioned experimental scenario. We plan to apply it in our further research. In the following section, we describe the model’s application scenarios and the design of a real-time detection system for testing the model.

#### 3.3.1. Sheep Behaviour Recognition Scenarios

To better apply the model and detect the behaviour of sheep in grazing scenarios, we place the camera above the water source for sheep behaviour recognition (Figure 11). We use video from the camera to observe the sheep’s state by detecting their daily behaviour remotely, such as feeding.

#### 3.3.2. Real-Time Recognition System Design

This real-time monitoring system is based on a streaming server for detecting sheep behaviour, which can be viewed in real time on the client side with statistical analysis of some information. The system is divided into three main parts, one division is the camera, the second division is the server, and the third division is the client (Figure 12). The effect of real-time recognition and viewing is achieved through video streaming services. The camera needs to support common push streaming protocols, including Real-Time Messaging Protocol (RTMP), etc.; the server side needs an intranet fixed internet protocol (IP) and a simple real-time server (Srs) as a streaming media server which supports common streaming media transmission protocols as well as pull and push streaming services. An improved YOLOV5 sheep behaviour recognition model is running on the server faction to change the input source to a streaming address. The recognised images are encapsulated into a video and pushed to the playback address; the client can view the recognised video in a web browser.

## 4. Discussion

When performing model training comparisons with photos from different shooting styles, the data from the two shooting styles differed; however, their label ratios were essentially the same. The models differed by only 0.6% on their respective validation sets’ mAP_@0.5_, while mAP_@0.5_ with models from different shooting styles across the entire dataset differed by less than 1%. Cheng et al. [16] investigated the performance of sheep behaviour in housing on different picture features, with various features based on different shooting angles to build features, pointing out that very few data are needed to achieve good results when the scene is fixed. Yang et al. [4] used a fully convolutional network (FCN) to segment images of lactating sows with different scenes, variable illumination, etc. When selecting data for the training model, there is a limit to the number of photos for the same scene. The model convergence cannot continue to improve accuracy after reaching a certain number of iterations. When performing behavioural recognition of sheep in grassland, multiple scenes of data collection are used to maintain the robustness of the model.

We selected the YOLOV5 model for improving the behavioural recognition model. As shown in Table 7, YOLOV5 outperforms other models with only small differences in recognition accuracy, but with fewer parameters and faster inference speed. Additionally, previous studies on digital agriculture have extensively used and applied YOLOV5, which has demonstrated both high accuracy and fast inference speed. Therefore, we chose to utilise and improve the YOLOV5 model for our study. Qi et al. [33] proposed an improved SE-YOLOV5 network model for identifying tomato virus diseases, which achieved a 1.87% increase in mAP_@0.5_ compared to the original YOLOV5 model. The SE-YOLOV5 model can effectively detect regions affected by tomato virus diseases. Wang et al. [34] proposed an enhanced YOLOV5-based method for detecting cow oestrus behaviour, which increased the model’s accuracy in identifying cow oestrus events by 5.9%. To improve the model, we explored the progress of the model accuracy by adding different modules. Hence, the recognition effect was improved by adding several modules. The final mAP_@0.5_ showed an improvement of 1.7%, as shown in Table 8. The results in Figure 13 demonstrate the successful performance of the model in recognising sheep behaviour. Nevertheless, obscuration hindered the accuracy of judging the behaviour of the sheep. For instance, if the sheep’s position was relatively close to the picture, the sheep was unidentifiable, resulting in low accuracy. Additionally, the angle of the camera view has a significant impact on the quality of the image, and it may be beneficial to use multiple cameras to capture different perspectives of the sheep’s behaviour and obtain more comprehensive data.

Real-time detection is a valuable application for models, as they can be integrated into real-time monitoring systems to maximise their benefits. This paper presents a design solution for a real-time system, providing guidance for model deployment. Wang et al. [35] developed the YOLO-CBAM convolutional neural network model to detect Solanum rostratum Dunal seedlings and conducted real-time testing using an inspection vehicle equipped with cameras, image processors, and other devices. Real-time detection systems can either rely on server-side detection through network transmission or deploy models on edge devices for detection, with the choice mainly determined by the specific detection scenario.

Since this study focuses on the daily behavioural recognition of sheep in pastures, in the future, more scenarios in sheep abnormality recognition and sheep social behaviour can be combined to ensure more sensitive and timely recognition of abnormal sheep (those exhibiting behavioural abnormalities such as movement disorders and aggressive behaviour) in the pasture.

## 5. Conclusions

From the focal point of sheep behaviour recognition in the pasture, this study utilised different shooting methods to model sheep’s daily behaviour (standing, feeding, lying) based on the YOLOV5 algorithm in the pasture terrain. It can be conclusively stated that identifying sheep behaviour using two angles is feasible, giving the model a certain generalisation ability. Based on the YOLOV5 algorithm, the model was improved. The improvement was achieved by adding the CBAM, BiFPN module, and SKNet-sc module to the model, resulting in model mAP_@0.5_ enhancement, generating the highest value of 0.918, an increase of 1.7%. The study enables livestock researchers to observe sheep behaviour visually and clearly, thereby saving time and labour in various studies. This advancement facilitates researchers in acquiring data more rapidly, holding significant implications for remote observation and diagnosis within the realm of precision farming. However, there are certain limitations to this study. Firstly, the study site is relatively small, and it is recommended that in a real pasture application, cameras should be positioned at water sources and specific locations such as fence boundaries to enable multi-angle detection. Additionally, the sheep involved in the study were limited to a single breed and grazing scenario. To address these limitations, we plan to expand our research in two key ways. Firstly, we aim to investigate the behaviour of sheep in multiple breeds and flocks. Secondly, we plan to incorporate additional behaviours such as running, walking, and social interactions into our video-based behavioural recognition system. These improvements will enhance the utility of our research findings and facilitate remote observation and diagnosis in the context of precision farming.

## Figures and Tables

**Figure 1 sensors-23-04752-f001:**
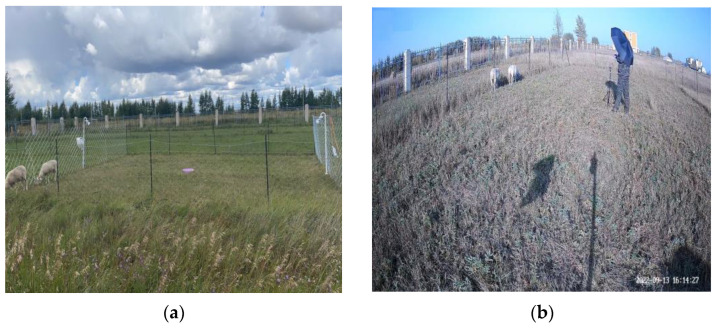
(**a**) Fixed cameras on both sides of the area. (**b**) The manual use of a motion camera for follow-along shooting.

**Figure 2 sensors-23-04752-f002:**
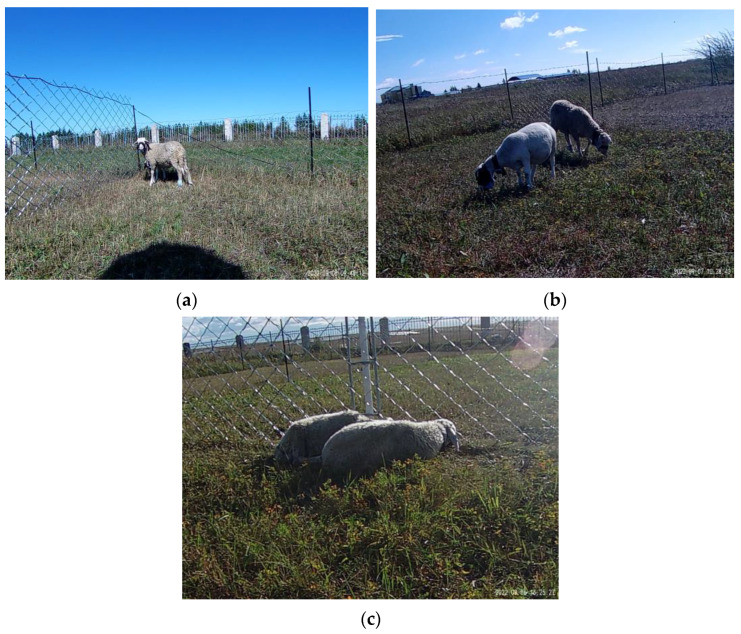
(**a**) Standing behaviour. (**b**) Feeding behaviour. (**c**) Lying behaviour.

**Figure 3 sensors-23-04752-f003:**
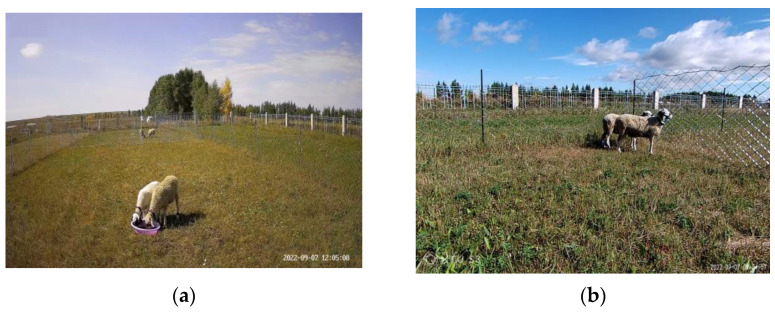
(**a**) Fixed camera pictures. (**b**) Handheld camera pictures.

**Figure 4 sensors-23-04752-f004:**
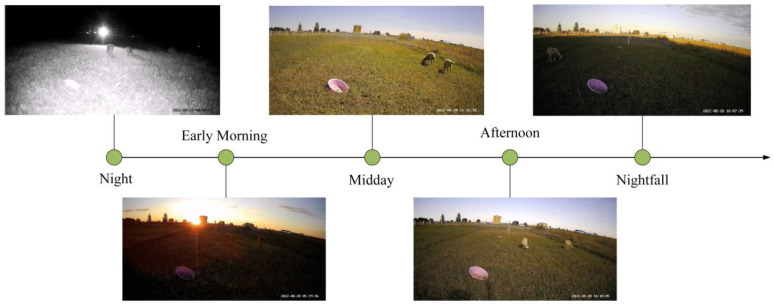
Images under different lighting conditions throughout the day.

**Figure 5 sensors-23-04752-f005:**
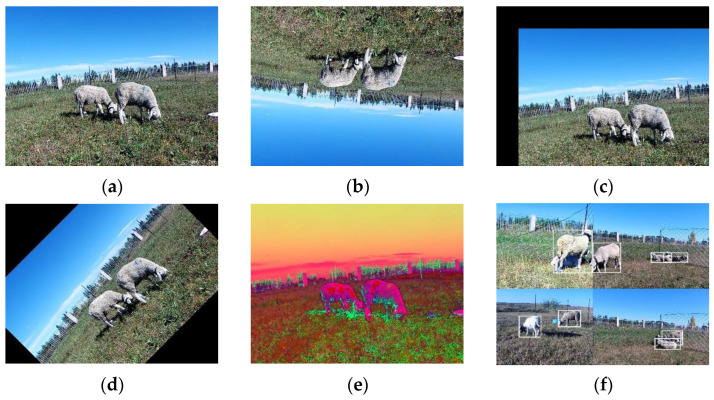
(**a**) Original image. (**b**) Image after flipping. (**c**) Image after panning. (**d**) Image after rotation. (**e**) Image after gamut change. (**f**) Mosaic data enhancement.

**Figure 6 sensors-23-04752-f006:**
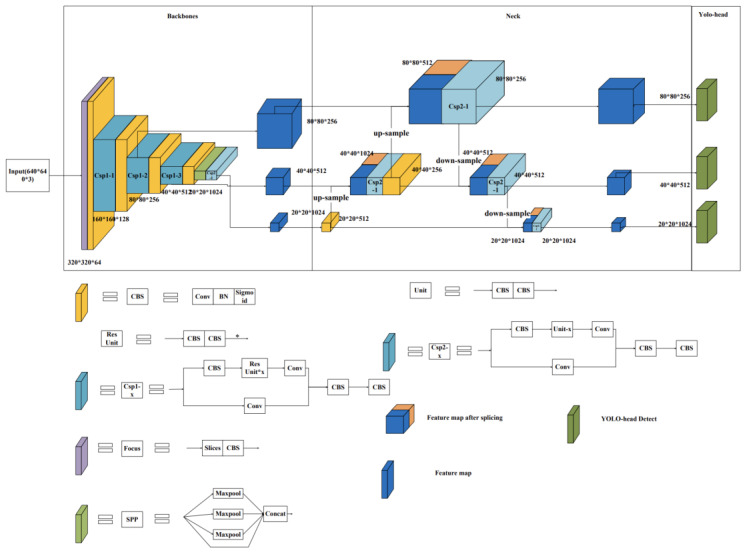
YOLOV5s network model structure diagram.

**Figure 7 sensors-23-04752-f007:**
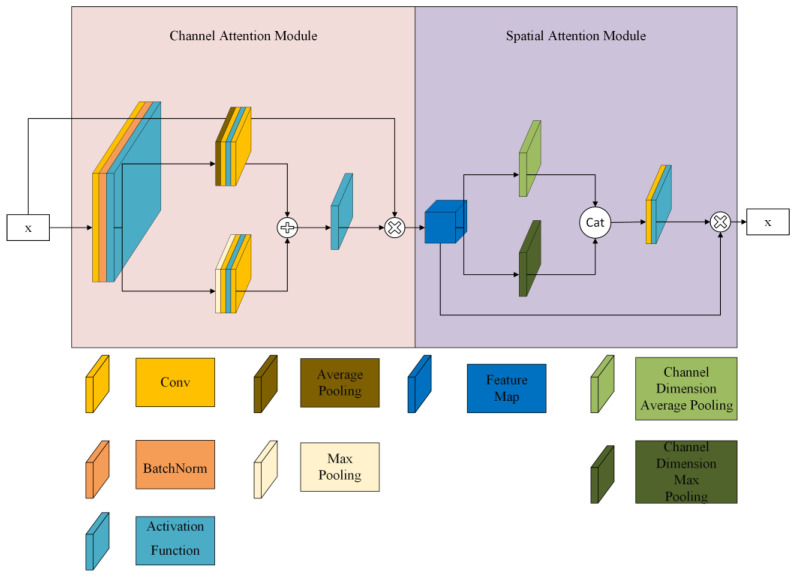
CBAM network structure diagram.

**Figure 8 sensors-23-04752-f008:**
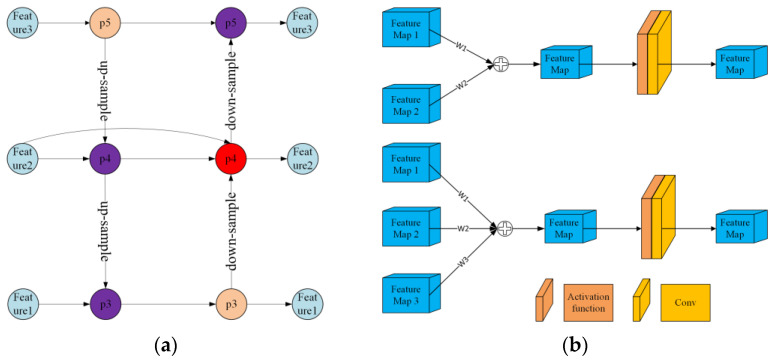
(**a**) Structure diagram of BiFPN in YOLOV5, where two features are fused, shown in purple, and three features are fused, shown in red. (**b**) Structure diagram of the feature fusion process, where w1, w2, and w3 are weights.

**Figure 9 sensors-23-04752-f009:**
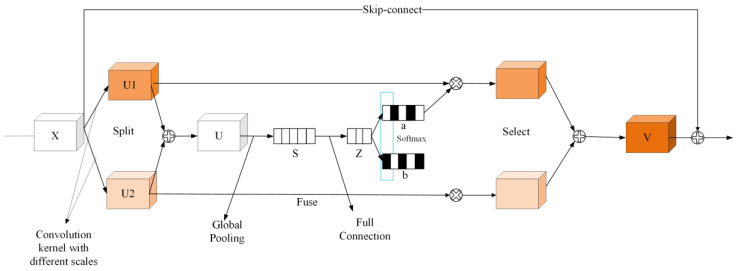
The network structure of the SKNet-sc module.

**Figure 10 sensors-23-04752-f010:**
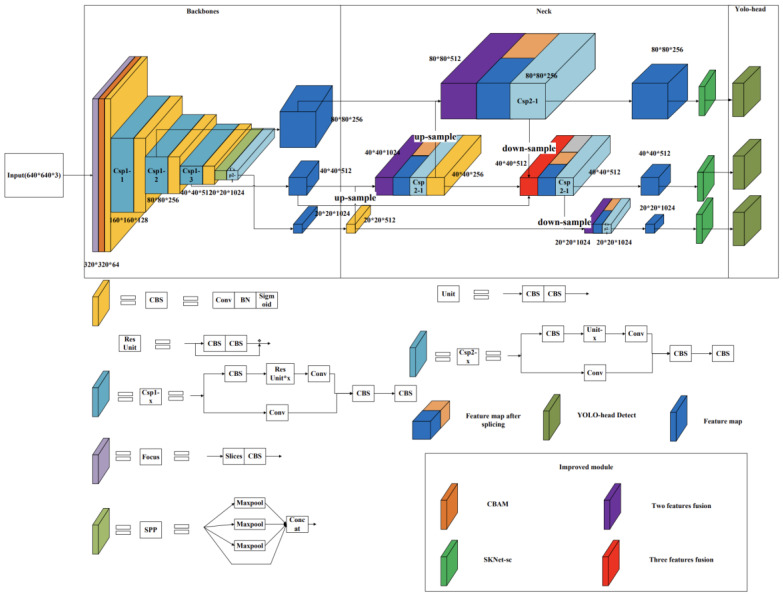
The improved network structure of the YOLOV5 model.

**Figure 11 sensors-23-04752-f011:**
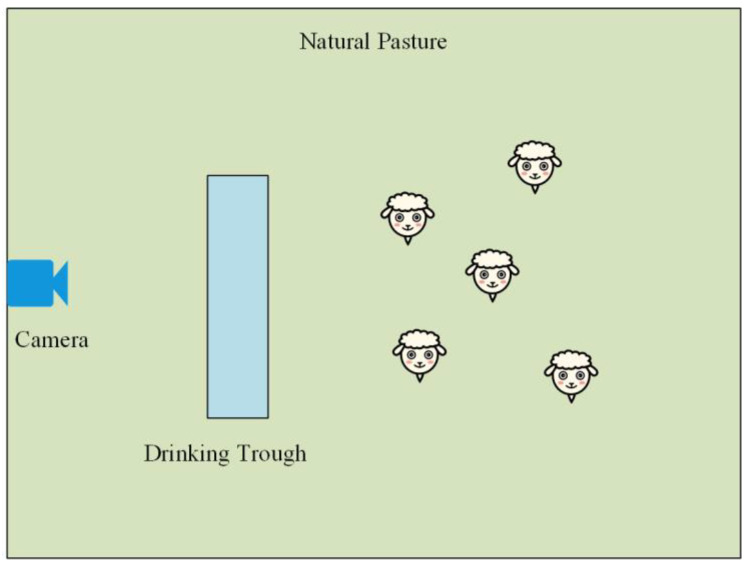
Sheep identification scene map.

**Figure 12 sensors-23-04752-f012:**
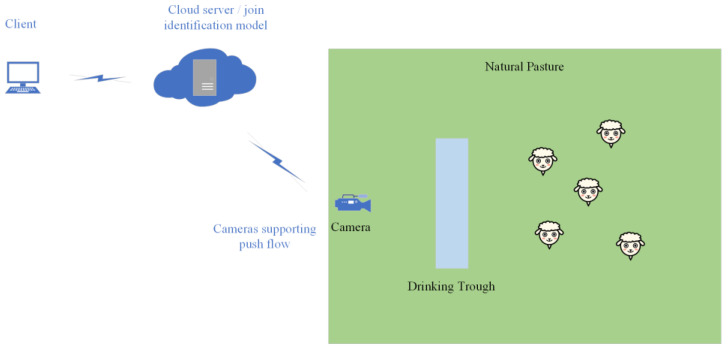
Real-time system application scenario diagram.

**Figure 13 sensors-23-04752-f013:**
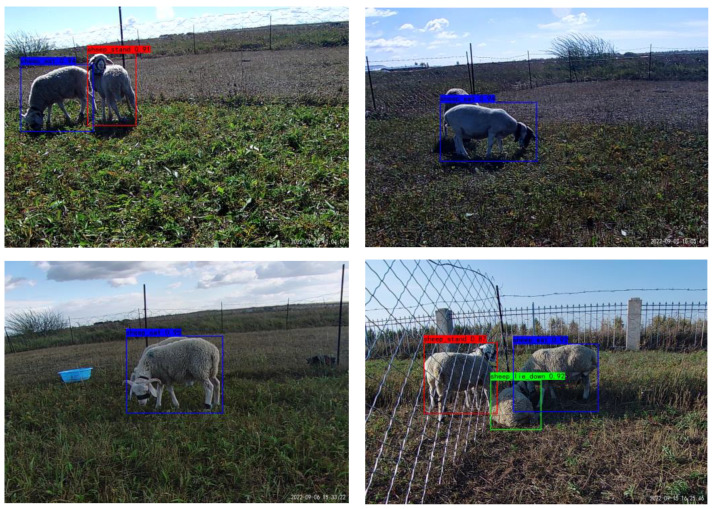
Optimal model prediction results.

**Table 1 sensors-23-04752-t001:** Definition of sheep behaviour.

Behaviour Name	Definition
Feeding	When the sheep’s head is positioned below the body in the picture
Standing	When the sheep’s head is positioned above or horizontal to the body in the picture
Lying	When the sheep’s body is on the ground in the picture

The table describes how we determine the behaviour of the sheep when selecting photographs.

**Table 2 sensors-23-04752-t002:** Overview of different scenario datasets.

Shooting Style	Overview of the Dataset
Fixed cameras	The dataset consists of 703 images, including 632 images in the training set and 71 images in the validation set, with 416 standing labels, 263 lying labels, and 936 foraging labels.
Handheld camera	The dataset consists of 953 images, with 857 images in the training set and 96 images in the validation set, with 685 standing labels, 337 lying labels, and 1016 foraging labels.

**Table 3 sensors-23-04752-t003:** Introduction to different scenario datasets.

Datasets	Features
Fixed cameras datasets	These pictures are categorised according to significant lighting influence, distinct variations between images, and a considerable range in the proportion of captured sheep within each image.
Handheld camera datasets	The pictures were taken under consistent lighting conditions, resulting in minimal lighting variation between them and providing clear and detailed captures of the sheep.

**Table 4 sensors-23-04752-t004:** Model evaluation results for different scenarios.

Filming Methods	Behaviour	P (%)	R (%)	F1	AP (%)	mAP_@0.5_ (%)
Fixed cameras	Feeding	92.4	91.2	0.92	94.6	91.1
	Standing	88.6	78.2	0.83	84.9	
	Lying	95.6	87.8	0.92	94.0	
Handheld cameras	Feeding	92.7	91.2	0.92	93.7	91.7
	Standing	91.5	84.8	0.88	90.1	
	Lying	93.4	88.5	0.91	91.3	

**Table 5 sensors-23-04752-t005:** Representation of Model cross-validation results for both scenarios.

Filming Methods	Validation Models	Behaviour	P (%)	R (%)	F1	AP (%)	mAP_@0.5_ (%)
Fixed cameras	Handheld camera dataset generated model	Feeding	91.3	89.2	0.90	91.6	83.5
		Standing	80.8	75.1	0.78	78.8	
		Lying	91.0	71.7	0.80	80.0	
Handheld cameras	Fixed camera dataset generated model	Feeding	74.2	87.5	0.80	84.3	74.2
		Standing	91.0	62.9	0.74	72.0	
		Lying	93.7	62.5	0.75	66.3	

**Table 6 sensors-23-04752-t006:** YOLOV5 training overall dataset results.

Behaviour	P (%)	R (%)	F1	AP (%)	mAP_@0.5_ (%)
Feeding	92.5	89.0	0.91	92.7	90.1
Standing	92.0	77.1	0.84	85.8	
Lying down	96.6	85.0	0.92	92.1	

**Table 7 sensors-23-04752-t007:** Comparison between YOLO versions.

Model	mAP_@0.5_ (%)	mAP_@0.75_ (%)	mAP_@0.5:0.95_ (%)	Model Size (MB)	FPS (Frame/s)
YOLOV5s	90.1	68.3	63.1	27.1	48
YOLOV6s	90.9	67.8	61.6	36.2	45
YOLOV7	91.3	66.2	61.0	71.3	38

**Table 8 sensors-23-04752-t008:** Results of ablation experiments.

YOLOV5s	CBAM	BiFPN	SKNet-sc	mAP_@0.5_ (%)	mAP_@0.75_ (%)	mAP_@0.5:0.95_ (%)	Model Size (MB)	FPS (Frame/s)
√				90.1	67.3	63.1	27.1	48
√	√			91.0	68.4	63.5	27.1	35
√		√		91.2	69.5	63.6	27.4	48
√			√	90.6	67.0	62.0	138.0	26
√	√	√		91.1	68.1	63.3	27.4	48
√	√		√	90.9	67.9	63.0	138	22
√		√	√	90.6	68.0	63.0	138.3	26
√	√	√	√	91.8	69.9	63.4	138.3	22

## Data Availability

Data available on request from the authors.

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
