# Peer review of "Grazing Sheep Behaviour Recognition Based on Improved YOLOV5"

_sensors, 2023, doi:10.3390/s23104752_

Round 1

Reviewer 1 Report

Aim of the study was to improve an already existing recognition algorithm in livestock behaviour observations. There are a few limitations of the study which are mentioned in the reviewd version attached.

Mostly, improving of the methodology description, discussions and conclusions is needed. Other questions/recommendations are mentioned in the text.

Only a few grammatical errors and typos were been found and indicated in the text.

Author Response

请参阅附件。

Reviewer 2 Report

The manuscript shows a good contribution to the field of Real-time recognition, it will be interesting to read future updates for detection at night.

Just check how the figures and tables will be named, whether they are all in bold or none.

Reviewer 3 Report

The authors of this manuscript used an improved YOLOV5 algorithm for grazing sheep behavior recognition, which had an average accuracy more than 90% and possessed importance for precision livestock management. This work is well organized and give a compelling conclusion. Thus, I recommended the acceptance of this study after minor revisions. Some issues can be discussed as below:

1.     Figure 2, 3, 12, 13 should be cited in the main text.

2.     The authors showed that YOLOV5 network model performed best among the YOLO family. Please give more discussion about the reason.

3.     The authors considered the different lighting conditions throughout one day, and I wonder that if other factors can influence the results of improved YOLOV5 model? For example, can it work well in different weather?

4.     The discussion part is not enough.

5.     In the conclusion part, the author may mention the limitation of the current study and the further improvement plan.

6.     There are some errors in the text. For example, in line 59, “[1,6,7,8,9,10,11]” should be revised as “[1,6-11]”. In line 84, “[24” should be revised as “[24]”. Please check the manuscript carefully.

English of this manuscript is OK. Some minor revisions are required.

Round 2

Reviewer 1 Report

Thank you very much for the revised version of the manuscript! It is significantly improved (not only by my recommendations) and it can be accepted in present form.

Hope your continous research work will signicantly affect the recognition of sheeps' (and other livestock, as well) behavioural elements on grasslands.